# Assessment of the Pharmacokinetics and Pharmacodynamics of Injectable Lidocaine and a Lidocaine-Impregnated Latex Band for Castration and Tail Docking in Lambs

**DOI:** 10.3390/ani14020255

**Published:** 2024-01-13

**Authors:** Joseph A. Ross, Steven M. Roche, Kendall Beaugrand, Crystal Schatz, Ann Hammad, Brenda J. Ralston, Andrea M. Hanson, Nicholas Allan, Merle Olson

**Affiliations:** 1Chinook Contract Research Inc., Airdrie, AB T4A 0C3, Canada; joe.ross@ccr01.com (J.A.R.); kendall@ccr01.com (K.B.); crystal.schatz@ccr01.com (C.S.); ann.hammad@ccr01.com (A.H.); nick.allan@ccr01.com (N.A.); 2ACER Consulting Ltd., Guelph, ON N1G 5L3, Canada; sroche@acerconsult.ca; 3Applied Research Team, Lakeland College, Vermilion, AB T9X 1K5, Canada; brenda.ralston@lakelandcollege.ca (B.J.R.); andrea.hanson@lakelandcollege.ca (A.M.H.); 4Alberta Veterinary Laboratories Ltd., Calgary, AB T2C 5N6, Canada

**Keywords:** effective concentration, anesthetic, elastrator, pain control, lidocaine

## Abstract

**Simple Summary:**

Tail docking and castration in lambs are common husbandry practices, both of which cause pain and discomfort, for which many industries recommend or require pain management. The objectives of this study were to assess the effects of the current standard-of-care for pain mitigation in lambs (injectable lidocaine) and assess the ability of a lidocaine-impregnated elastration ligation band to deliver the drug into the contacted tissues over time. The use of injectable lidocaine provides effective short-term anesthesia for 120 to 180 min following the injection; however, additional strategies are needed to manage long-term pain. The use of a ligation band impregnated with lidocaine could provide a useful alternative, as it appears to offer local anesthesia for at least 3 days when compared to a control band. Further studies are needed to compare the use of an injectable local anesthetic to the Lidocaine-Loaded Bands (LLBs).

**Abstract:**

The objectives of this study were to assess the pharmacokinetics and pharmacodynamics of the current standard-of-care for pain mitigation in lambs during castration and tail docking (injectable lidocaine) and assess the ability of Lidocaine-Loaded Bands (LLBs) to deliver therapeutic concentrations into the contacted tissues over time. The study was comprised of four different trials: (1) investigation of in vitro release of lidocaine from LLBs; (2) pharmacokinetics and pharmacodynamics of injectable lidocaine in scrotal and tail tissue; (3) pharmacokinetics and pharmacodynamics of in vivo delivery of lidocaine with LLBs placed on the tail and scrotum of lambs; and (4) a “proof-of-concept” study comparing the sensation of control- versus LLB-banded tail tissue over time. The use of injectable lidocaine provides effective short-term anesthesia for 120 to 180 min following the injection; however, additional strategies are needed to manage long-term pain. The use of an LLB could provide an alternative where tissue lidocaine concentrations meet or exceed the EC_50_ for at least 21–28 days and, based on electrostimulation data, provides local anesthesia for at least 3 days when compared to a control band. Further studies are needed to compare the use of an injectable local anesthetic to the LLBs.

## 1. Introduction

Tail docking and castration are commonly performed on lambs. Tail docking is completed to manage fecal soiling, which is associated with the development of myiasis or fly strike [1]. Castration is commonly practiced in ram lambs to eliminate the sexual behavior of young males and reduce aggression, as well as prevent unwanted pregnancies and indiscriminate breeding [2,3]. Further, there is no risk of ram taint in rams that have been castrated, which improves meat quality [4].

No matter the method used to perform tail docking and castration, these procedures are painful, as they stimulate nociceptors by causing tissue damage and triggering physiological pain pathways [5]. As a result, lambs commonly exhibit pain-associated behaviors when these procedures are being conducted, such as vocalization, tail wagging, and restlessness [6]. Elastomeric rings are commonly used to perform these procedures due to their ease of application; however, these tight rubber rings prevent blood supply to the tail and do not prevent the conduction of nerve impulses from the painful, ischemic tail, causing a protracted pain response [7,8].

To combat the pain associated with the application of a rubber ring around the tail and scrotum, a multi-modal approach should be used. Local anesthetic administered into the scrotal neck and cord for castration can lead to reduced cortisol responses and behavioral indicators of pain and discomfort in the period immediately following the application of the band [8,9]. For tail docking, using a local anesthetic at the site of application or giving an epidural prior to the application reduces cortisol responses and pain behavior in the hours following application [8,10]. With regard to non-steroidal anti-inflammatory drugs (NSAIDs), fewer studies have been completed. One study found that the injection of an NSAID reduced cortisol and behavioral responses associated with pain in the hours following castration [11]; however, others have found few differences upon application at castration on acute pain behavior [10] or a slightly reduced average daily gain but lower level of lamb losses [12]. Graham et al. [13] found that the injection of an NSAID following tail docking reduced cortisol response but not behavioral indicators of pain, whereas Pollard et al. [14] found the oral administration of an NSAID reduced restlessness, active behavior, and abnormal posture following tail docking. In any case, NSAIDs would require repeated re-administration to maintain efficacy over a period of days to weeks, which is likely not practical. Despite the utility of a multi-modal approach for pain control, the use of rubber rings can create long-term pain, as it can take considerable time for the procedure to be completed; for example, castration takes more than 4 weeks following the application of the rubber ring [15]. Given that the scrotum continues to cause behavioral signs of discomfort prior to casting (i.e., detachment), the development of longer-term pain control strategies is needed.

Effective concentrations of local anesthetics, such as the tissue concentration yielding a 50% and 95% reduction in tissue sensation (EC_50_ and EC_95_, respectively), are important metrics of an anesthetic’s potency [16,17,18]. Moreover, by measuring the tissue concentration over time and comparing it to the EC_50_, the time of onset and duration of local anesthesia can be established [19,20]. Lidocaine is a well-studied local anesthetic in humans [21,22,23]. However, despite a few studies looking at the systemic pharmacokinetics of lidocaine in sheep and some other species [24,25,26], there is a paucity of evidence with regard to lidocaine’s pharmacokinetics, pharmacodynamics, and effective concentrations for local anesthesia in scrotal or tail tissue. Given the widespread recommendation—and, in many cases, requirements (see Canadian Codes of Practice for the Care and Handling of different livestock species [27])—for the use of pain control (often citing the importance of local anesthetics), more research is clearly needed to provide a more comprehensive understanding of lidocaine use in these species.

A latex elastration device has been developed that has lidocaine impregnated directly into the band (US patent #11596510 assigned to Chinook Contract Research Inc.) [28]. This may allow for the slow release of lidocaine into the site of application. In cattle, it has been found that these Lidocaine-Loaded Bands (hereafter, LLBs) had similar short-term performance to lidocaine injections; however, LLBs outperformed the standard injections for long-term delivery of lidocaine and pain mitigation [28]. Despite the work in cattle, no studies have been completed on LLBs in lambs. Hence, the objectives of this study were to assess the pharmacokinetics and pharmacodynamics of the current standard-of-care for pain mitigation in lambs during castration and tail docking (injectable lidocaine) and assess the ability of LLBs to deliver therapeutic concentrations into the contacted tissues over time.

## 2. Materials and Methods

These studies, when animals were used, were conducted in compliance with the animal care guidelines established by the Canadian Council of Animal Care and were approved by the Chinook Contract Research IACUC (A8217-03). The study was comprised of four different trials: (1) investigation of in vitro release of lidocaine from LLBs; (2) pharmacokinetics and pharmacodynamics of injectable lidocaine in scrotal and tail tissue; (3) pharmacokinetics and pharmacodynamics of in vivo delivery of lidocaine with LLBs placed on the tail and scrotum of lambs; and (4) a “proof-of-concept” study comparing the sensation of control- versus LLB-banded tail tissue over time. In studies using the LLBs, the latex rubber band contained 80 mg of lidocaine base USP (Lidoband^TM^, Solvet, AB, Canada). All study lambs were returned to their home pen with their dam, where they were housed and fed in a normal commercial setting. The meat withdrawal time for lidocaine for livestock in Canada is 5 days following the last treatment; no animals were slaughtered within this lidocaine withdrawal period.

### 2.1. In Vitro Lidocaine Release from LLBs

To analyze the rate of lidocaine being released from the LLBs, dissolution testing was conducted. One LLB was placed into each of six baskets, and the baskets lowered into vessels containing 900 mL of PBS at 35 ± 1 °C (baskets spinning at 100 rpm) to initiate the time course. Samples (1 mL) were removed at T = 0 (i.e., prior to lowering the baskets), 30 min, 1 h, 2 h, 6 h, 24 h, 48 h, and 168 h (7 d) for analysis of lidocaine content by UV-Vis spectroscopy (using Absorbance at 263 nm). The assay used here was based on USP <711> (Dissolution), USP <724> (Drug Release), and USP <1092> (The Dissolution Procedure: Development and Validation). Data were best fit with an equation for 2-phase release.

### 2.2. Pharmacokinetics, Pharmacodynamics, and Effective Concentrations (EC_50_, EC_95_) of Injectable Lidocaine in Lamb Scrotal and Tail Tissue

To understand the pharmacokinetics, pharmacodynamics, and effective concentrations of injectable lidocaine-HCl, 6 intact male lambs (<1 month of age) were enrolled at a commercial farm near Airdrie, AB, Canada. The lambs were randomly selected by the producer and not according to a computer-generated random number. To participate in the study, the lambs had to be healthy based on a general physical examination prior to enrollment and could not have been treated with lidocaine or NSAIDs previously. Once enrolled, six injection sites (on each animal, at both the scrotum neck and distal end of the caudal fold of the tail) were marked with a 1 cm circle over the scrotal surface with a permanent marker (Figure 1).

The animals were then injected with 2% lidocaine-HCl without epinephrine (Teligent Canada Inc., Mississauga, ON, Canada) into scrotal neck tissue and tail tissue at a dose of 1 mL at each scrotal injection site or 0.5 mL at each tail site, forming a “ring block”. Ring blocks are a recommended procedure to provide local anesthesia for castration and tail docking of lambs (NFACC, 2013). Locations of the injection sites were randomized for each animal. At times 30, 60, 90, 120, 180, and 240 min after injection, a punch biopsy (Acu-Punch, 4 mm diameter, 7 mm depth; Acuderm Inc., Ft. Lauderdale, FL, USA), comprised of both skin and subcutaneous tissue, was collected from the center of the injection site area (as defined by the colored ink circle) at both scrotal and tail sites. Samples were collected from all 6 animals at each time point at one of the injection sites selected according to the randomized sampling schedule. Two additional animals served as negative controls; these underwent knife castration, with biopsy samples taken to provide lidocaine-free control tissue. All animals were then treated with oral meloxicam (1 mg/kg body weight; Solvet, Calgary, AB, Canada) and observed for adverse events for up to 72 h following study completion (none were observed). The local anesthetic activity was determined by grading the reflex response associated with electrocutaneous stimulation, as previously demonstrated [29]. The injection site, defined by the colored ink circle, was stimulated by applying infant monitoring electrodes (Red Dot 2258, 3M Canada, London, ON, Canada, or similar) over the identified site. Electrodes were attached to the peripheral nerve stimulator so that the lambs could be stimulated without being restrained. This was done before the biopsy times at the same designated circle. The Sun Stim Peripheral Nerve Stimulator (SunMed, Largo, FL, USA) frequency was 100 Hz, and the output current was adjustable from 0 to 250 mA. For baseline and post-treatment administration, both the scrotal and tail tissues were stimulated at five increasing stimulus levels (median current; range (mAmp)), 2 (<30), 4 (90; 87–95), 6 (160; 155–162), 8 (219; 211–225), and 10 (244; 243–245), until a positive avoidance response was observed and graded as at least a 2 (Table 1).

### 2.3. In Vivo Delivery of Lidocaine from LLBs into Scrotal and Tail Tissues

Intact male lambs were selected from a commercial flock near Airdrie, AB, Canada. To participate in the study, the lambs had to be healthy based on a general physical examination prior to enrollment and could not have been treated with lidocaine or NSAIDs. Cryptorchid lambs and lambs with inguinal hernias were not included in the study. Field technicians were masked as to the identity of control versus test bands. At time 0 (T = 0), a banding tool was used to place an LLB around both the neck of the scrotum and the distal end of the caudal fold of the tail (one band per site; two sites per animal) of 45 male lambs. A different group of 5 animals were tested at each time point (0.5 h, 1 h, 2 h, 24 h, 72 h, 3 d, 7 d, 14 d, 21 d, and 28 d after banding). Animals were randomly assigned to their treatment group by treating with either control bands or LLBs according to a random blocking factor; to ensure a similar average starting body weight for each group, the animals were weighed and ranked heaviest to lightest prior to applying the blocking factor.

The in vivo release rate and amount of lidocaine at each time-point were measured by removing bands and collecting two 4 mm punch tissue biopsies (containing skin and subcutaneous tissue) from the area that had been in direct contact with the band, with one of these serving as a backup sample. Each biopsy sample was placed in a separate microtube and frozen, then stored at −80 °C for processing at a later point. A group of 5 control intact males were also tested.

### 2.4. Assessment of Tissue Lidocaine Concentrations

The analysts were blinded to the animal treatment group and sampling time. Biopsy samples were stored at −80 °C until processing. Samples were thawed and placed into pre-weighed tubes containing 2.8 mm ceramic beads (Cat # 10158-612, VWR, Mississauga, ON, Canada) and 1 mL of “mobile phase” (40:60 acetonitrile:Phosphate Buffered Saline, pH 7.4). Tubes were weighed to calculate the net weight of tissue and subjected to homogenization in a FisherBrand Bead Mill 24 Homogenizer (Model No. 19-2241A) as follows: 2 cycles (5 min per round) at 6 m/s, 20 °C, with >1 min of cool-down time between rounds. Tubes were then centrifuged (5 min at 12,000× *g*) to pellet cell debris, and 800 µL of supernatant was removed and passed through a nylon filter (0.45 µm pore size). The samples were then analyzed for lidocaine content by High-Performance Liquid Chromatography (HPLC) using the parameters outlined in Table 2. Lidocaine content (mg lidocaine per mL of mobile phase) was calculated from a standard curve and converted to mg lidocaine per g (wet weight) of tissue.

### 2.5. Assessment of Local Anesthesia in Control versus LLB-Banded Tails

Intact male lambs were selected from a commercial flock near Strathmore, AB, Canada. To participate in the study, the lambs had to be healthy based on a general physical examination prior to enrollment and could not have been treated with lidocaine or NSAIDs previously. Cryptorchid lambs and lambs with inguinal hernias were not included in the study. At time 0 (T = 0), a banding tool was used to place a control band or an LLB around the distal end of the caudal fold of the tail. The local anesthetic activity was determined by grading the reflex response associated with electrocutaneous stimulation as described above. Using preliminary data obtained during the development of the LLB used in this study, and assuming a delta of 1.0 for graded electrocutaneous stimulation response, statistical power of 80%, and an alpha of 0.05, a sample size of 11 animals per treatment group was calculated. A total of 12 animals per treatment group were enrolled, for a total of 24 lambs. For each animal, readings were taken just prior to band placement and at T = 1 h, 2 h, 4 h, 24 h, 48 h, 72 h, 7 d, 14 d, 21 d, and 28 d after banding. As a positive control for sensation in un-banded tissue, the electrodes were also placed approximately 5 cm above (i.e., proximal to) the band for a second reading at each time point. Note that due to the convenience of accessing the tail site relative to the scrotal site and due to the larger number of animals tested here relative to the previous electrostimulation study conducted in the absence of bands, this proof-of-concept study was only conducted on the tails. This had the added advantage of minimizing handling time and reducing stress for the animals.

### 2.6. Statistical Analysis

Descriptive statistics (mean, standard error on the mean) were generated for each of the studies described above. Non-linear regression was used to calculate the in vitro release rate of lidocaine from LLBs. For the injectable lidocaine study, a repeated measures one-way ANOVA was used to evaluate tissue lidocaine concentrations and electrostimulation response scores at each time point relative to the baseline sample; a correction for multiple comparisons was made using Dunnett’s test. Non-linear regression was also used to calculate the EC_50_ and EC_95_ values for lamb scrotal and tail tissues. For the in vivo delivery of lidocaine from LLBs, a one-way ANOVA was used to evaluate tissue lidocaine concentrations at each time point and was corrected for multiple comparisons using Dunnett’s test. Electrocutaneous response scores were analyzed with a repeated measures mixed-effects regression model. Fixed effects included within the model included time, treatment, and a time–treatment interaction, while animal was included as a random effect. *p*-values for individual time points were not corrected for multiple comparisons (i.e., Fisher’s LSD test was used). A significance level of 0.05 was used to identify meaningful differences between comparison groups.

## 3. Results

### 3.1. In Vitro Lidocaine Release from LLBs

LLBs were designed to deliver lidocaine rapidly (to address acute discomfort) and for a prolonged duration (to address chronic discomfort). In order to ensure that the LLBs performed as expected prior to moving on to in vivo testing, a dissolution experiment using a “basket” apparatus (i.e., Apparatus 1, per USP <711>) was used to quantitate in vitro release of lidocaine into an aqueous medium over a 1-week time course. Indeed, lidocaine release was best fit with a two-phase equation: release was initially rapid (K_obs_ = 2.98 mg/h) for the first 30.4 h of the time course, slowing to 0.0292 mg/h for the remainder (Figure 2). Lidocaine release approached a plateau of approximately 79.6 mg/band over the 1-week time-course (Figure 2).

### 3.2. Determination of Pharmacokinetics, Pharmacodynamics, and Effective Concentrations of Injectable Lidocaine

Animals displayed minimal reaction to lidocaine injections (i.e., minimal evasive behavior or vocalization. Lidocaine was undetected in all tissues at T = 0 (i.e., prior to lidocaine injection), reaching approximately 2.0 mg/g and 0.7 mg/g by 30 min post-injection in scrotal and tail tissues, respectively, and dropping back toward zero over the time-course (Figure 3A,B).

For all tested tissues, the electrostimulation response score was near maximum at T = 0 (i.e., prior to lidocaine injection), dropping to 0 for all animals by 30 min post-injection (Figure 3C,D), indicating a complete loss of sensation in the injected tissues. The response score then gradually rebounded to levels that were not significantly different (*p* > 0.05) from the T = 0 level between 120 and 180 min post-injection (Figure 3C,D), indicating a return of sensation.

The electrostimulation response scores from Figure 3C,D were plotted against the tissue lidocaine concentrations from Figure 3A,B, and non-linear regression was used to calculate the EC_50_ and EC_95_ values. EC_50_ was 0.174 and 0.0765 mg/g for scrotums and tails, respectively; EC_95_ was 2.08 and 0.608 mg/g for scrotums and tails, respectively (Figure 3E,F).

### 3.3. In Vivo Delivery of Lidocaine from LLBs into Scrotal and Tail Tissues

Lidocaine was not detected in tissue from control animals. In lambs treated with LLBs, tissue lidocaine levels reached or exceeded the EC_50_ in as little as 30 min (Figure 4A,B). Moreover, tissue lidocaine accumulated in these tissues, yielding a significant linear trend (*p* = 0.0081 and 0.007 for scrotums and tails, respectively; Figure 4A,B). Finally, lidocaine levels remained well above the EC_50_ for 21–28 days for tails and scrotums (Figure 4C,D).

### 3.4. Local Anesthesia in Control versus LLB-Banded Lambs

Finally, the response to electrostimulation at the tail site was compared for control- and LLB-banded animals. When tissue proximal to the band was stimulated, response scores were similar over time regardless of treatment group (Figure 5A), confirming that un-banded tissue had full sensation over the time course. On the other hand, when tissue in contact with the bands was stimulated, the response scores decreased over time regardless of treatment group (Figure 5B), indicating a loss of sensation in banded tissue over the time course; an exception was apparent at Day 7 when response scores transiently increased before resuming their downward trajectory. However, the LLB-banded group had a lower response score than the control group on days 1 and 3 following the application of the bands (Figure 5B).

## 4. Discussion

Given our collective knowledge of the pain associated with common procedures, including tail docking and castration in lambs, industries around the world have recommended the use of multi-modal pain management strategies. While lidocaine is a well-studied local anesthetic in humans [21,22,23], there is a paucity of evidence with regard to lidocaine’s pharmacokinetics, pharmacodynamics, and effective concentrations for local anesthesia in scrotal or tail tissue, information critical to guiding recommended dose rates and regimens.

Importantly, recent reviews of the literature on analgesia in sheep conclude that even with the use of a traditional local anesthetic and non-steroidal anti-inflammatory drugs, pain is not entirely ameliorated [30]. It is, therefore, critical that innovative strategies for pain management are investigated and developed. The latex elastration device impregnated with lidocaine used in this study [28] was designed to help facilitate improved and practical pain mitigation on farms. The objectives of this study were to assess the pharmacokinetics and pharmacodynamics of the current standard-of-care for pain mitigation in lambs during castration and tail docking (injectable lidocaine) and assess the ability of LLBs to deliver therapeutic concentrations into the contacted tissues over time.

From the evaluation of the current standard practice used to mitigate short-term pain (lidocaine injection), it is evident that providing a local anesthetic mitigates sensation for a brief period following administration. This is similar to what others have found where administration of a local anesthetic into the scrotal neck and tail led to reduced behavioral indicators of pain and discomfort [8]. With regard to tissue concentrations of lidocaine when injected, at 30 min, they reached about 2.0 mg/g and 0.7 mg/g in scrotal and tail tissues, respectively; however, both dropped toward zero close to 120 min after administration. Furthermore, the injected lidocaine was only able to mitigate response to electrostimulation for 120 to 180 min. This finding is similar to Mellema et al. [31] and Stewart et al. [32], who reported reductions in pain-related behaviors in the first 2- and 3-h post-procedure, respectively. Other studies have tended to suggest an effective duration between 1 and 2 h [33,34,35]. However, other studies have noted shorter effective durations. For example, Small et al. [26] reported that the duration of a number of local anesthetics is short in sheep, especially when lidocaine is used, after finding the period of significant effect was only the first 10 min post-procedure (though a tendency to reduce pain-related behaviors was observed over the first 60 min period). Crucially, differences reported between these studies often relate to dosage and site of administration. These and other reviews of the scientific literature [15,29] reiterate that a long-term pain control strategy is needed to mitigate pain associated with these procedures.

While a few studies have investigated the systemic pharmacokinetics of lidocaine in sheep [24,25,29], to the best of the authors’ knowledge, this is the first study to measure the pharmacokinetics, pharmacodynamics, and effective concentrations of injectable lidocaine in the tail and scrotum of lamb. The present study results demonstrate that EC_50_ was about 0.174 and 0.077 mg/g for scrotums and tails, respectively, whereas the EC_95_ was about 2.08 and 0.61 mg/g for scrotums and tails, respectively. These data suggest that more lidocaine is required to anesthetize scrotums than tails.

In the dissolution study, the test LLBs began to release lidocaine at 30 min, beginning to plateau after 48 h and releasing lidocaine at least until 168 h. Note that the dissolution assay, as employed here, is typically employed in the pharmaceutical industry to measure in vitro performance for purposes of quality control but is not necessarily intended to reflect in vivo performance. However, similar results were found by Saville et al. [28], who used a similar LLB to evaluate the transfer kinetics of lidocaine into ex vivo tissue (beef steak). In their study, lidocaine delivery by LLBs into tissue was initially fast, followed by a sustained delivery lasting at least 48 h. This coincides with the present in vivo study, where tissue concentrations of lidocaine began to increase between 0.5 and 24 h after band placement and remained above the EC_50_ until at least 21 days and 28 days for tail and scrotal tissues, respectively. This could lead to benefits for both acute and chronic pain management during castration or tail docking, with fast-acting pain relief at the time of band application and sustained pain relief over the course of the castration or tail-docking process. 

As proof of this concept, the electrostimulation response score of tail tissue was monitored over time after administration of LLBs versus control rings, indicating a general decline in response scores over time regardless of treatment group, which was likely due to the loss of sensation associated with the interruption of blood supply and subsequent ischemia of the tissue. Interestingly, a transient increase in electrostimulation response was observed at day 7 regardless of treatment group, which corresponds to the time at which the bands began to cut into underlying tissue. Notably, the tissue surrounding the band became inflamed at this time point, making it difficult to directly stimulate the tissue in direct contact with the band without also stimulating the surrounding inflamed tissue. Hence, future studies aiming to assess electrostimulation response past day 7 post-banding may require the development of electrostimulation probes that can better focus the stimulation on a selected area. Nonetheless, significant differences were observed in the response score on days 1 and 3 following the application of LLBs versus control bands. Stewart et al. [32] completed a similar study, finding a lower level of behavioral responses and cortisol in lidocaine-coated bands compared to conventional bands used in castrating lambs. However, the effects of the coated band were not as effective compared to an injected local anesthetic. In this study, a positive control group, or the application of a local anesthetic, was not used, and it should be encouraged in future studies. However, inferences could be made from the data presented in this and other studies where the injection of a local anesthetic has a short duration of activity: 120 min in this study and 30 to 180 min in other studies castrating lambs and beef cattle [26,36,37]. Hence, based on the electrostimulation data, it is highly probable that the LLB has a longer duration of anesthesia compared to injectable lidocaine; however, additional work is needed to verify these findings.

## 5. Conclusions

This study defined the effective concentrations of injectable lidocaine, yielding 50% or 95% reductions in local sensation (EC_50_ and EC_95_, respectively). The use of injectable lidocaine could provide effective short-term anesthesia for 120 to 180 min following the injection; however, additional strategies are needed to manage long-term pain. The use of an LLB could provide an alternative where tissue lidocaine concentrations meet or exceed the EC_95_ for at least 21–28 days and, based on electrostimulation data, provides local anesthesia for at least 3 days when compared to a control band. Further field and laboratory studies into LLB efficacy are ongoing, including a comparison of the use of an injectable local anesthetic to the LLBs.

## Figures and Tables

**Figure 1 animals-14-00255-f001:**
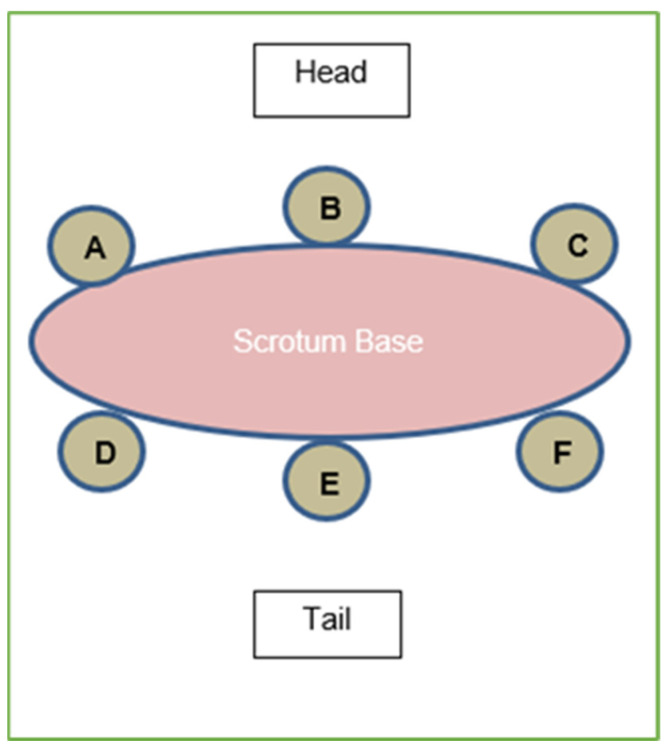
Sample injection and biopsy site layout for lamb scrotums. The same layout was used for lamb tails. Locations of sites A–F were randomized for each animal and were color-coded with markers. Treated lambs received an injection of 2% lidocaine (without epinephrine) at sites A–F to create a ring block; 1 mL for scrotal site, 0.5 mL for tail sites.

**Figure 2 animals-14-00255-f002:**
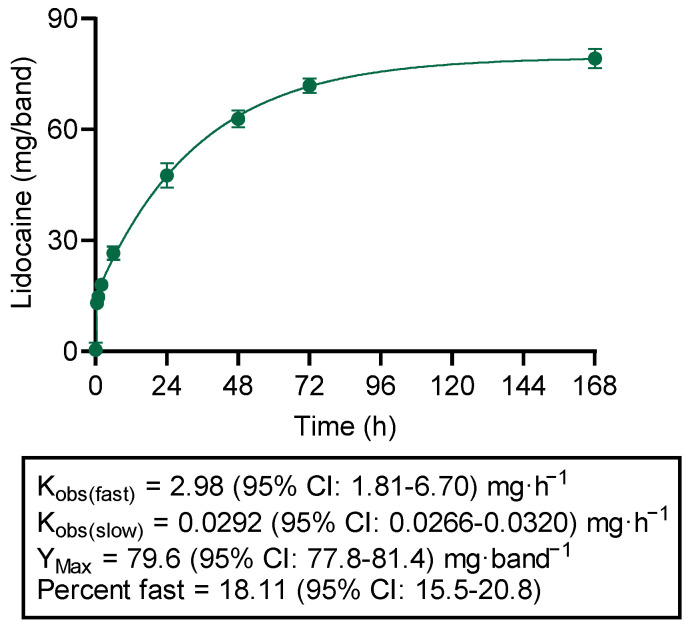
In Vitro Release of Lidocaine from Lidocaine-Loaded Bands (LLBs) (Dissolution Testing). Data represent the mean ± SD of 6 individual bands (i.e., 6 dissolution vessels). PBS was used as the dissolution medium. Data were best fit with an equation for 2-phase exponential release.

**Figure 3 animals-14-00255-f003:**
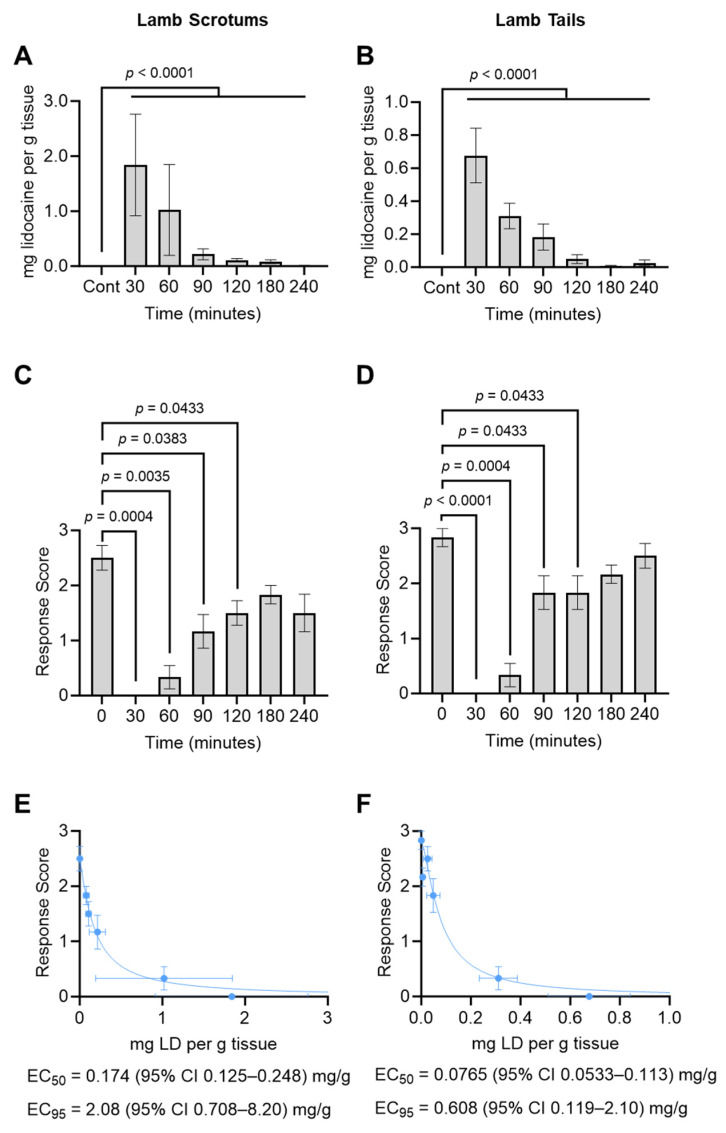
Pharmacokinetics/Pharmacodynamics of Injected Lidocaine. (**A**,**B**) Tissue lidocaine concentration for lamb scrotums (**A**) or lamb tails (**B**) at the indicated times after lidocaine injection (T = 0). (**C**,**D**) Electrocutaneous stimulation response scores, per Table 2, for lamb scrotums (**c**) or lamb tails (**D**) at the indicated times after lidocaine injection (T = 0). *p*-values were determined for each time-point relative to the T = 0 control sample using repeated measures one-way ANOVA and corrected for multiple comparisons using Dunnett’s test. Bars represent the mean ± SEM for 6 animals. (**E**,**F**) Stimulation response scores from panels (**C**,**D**) were plotted (on the *y*-axis) versus tissue lidocaine concentrations from (**A**,**B**) (on the *x*-axis), and non-linear regression was used to calculate the EC_50_ and EC_95_ values for lamb scrotal (**E**) or lamb tail (**F**) tissues.

**Figure 4 animals-14-00255-f004:**
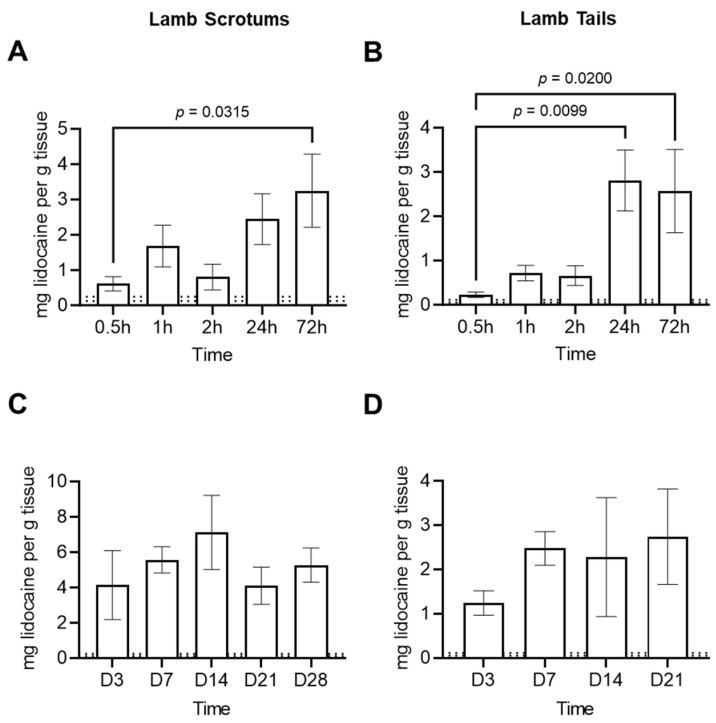
Lidocaine levels in lamb scrotal (**A,C**) or lamb tail (**B**,**D**) tissues biopsied at the indicated times after banding with Lidocaine-Loaded Bands (LLBs). For lambs, acute time-points (**A**,**B**) were assessed in an initial study, followed by chronic time points (**C**,**D**) in a follow-up study. In all cases, bars represent the mean ± SEM of 5 animals (note that a different set of animals was tested for each time-point). *p*-values (relative to the earliest time point) were determined with a one-way ANOVA and corrected for multiple comparisons using Dunnett’s test. For reference, the dotted lines denote the 95% CI of the EC_50_ (see Figure 3). No lidocaine was detected in tissue from control-banded animals (not shown).

**Figure 5 animals-14-00255-f005:**
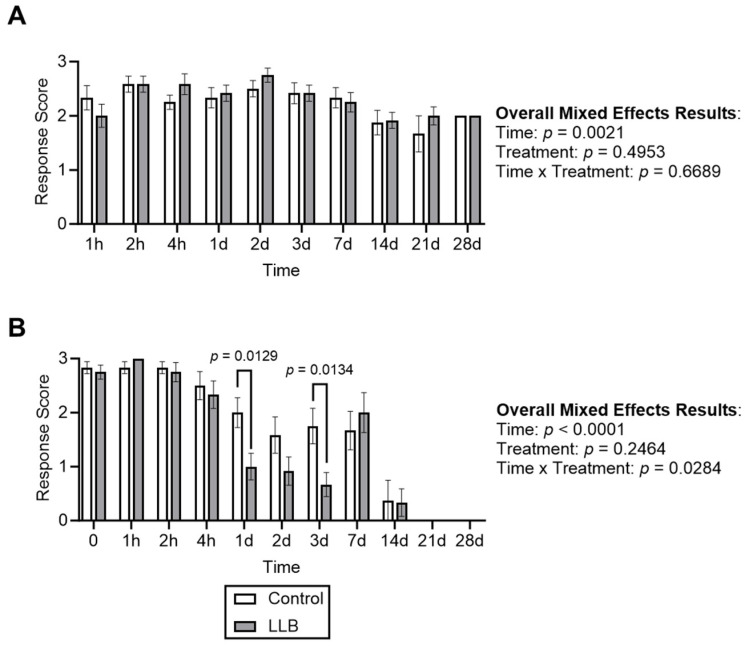
Electrocutaneous stimulation response scores over time for lamb tails treated with control bands or Lidocaine-Loaded Bands (LLBs). (**A**) As an overall positive control for sensory response, electrocutaneous response scores were taken above the band (rheostat set to 8, corresponding to 219 mA for all animals). The response scores at the band are presented in panel (**B**). For panel (**B**), the lowest rheostat level, giving the highest baseline score for each individual animal at T = 0, was used for all subsequent time points. Bars represent the mean ± SEM of 12 control-treated or 12 LLB-treated animals. Data were analyzed with a repeated measures mixed-effects model. *p*-values for individual time points were not corrected for multiple comparisons (i.e., Fisher’s LSD test was used).

**Table 1 animals-14-00255-t001:** Electrocutaneous stimulation rubric for sensitivity at biopsy location.

Graded Response	Description of Positive Avoidance Response
0	No reaction
1	Slight reaction: Moves side to side and tail flick
2	Moderate reaction: Moves side to side and tail flick, slight kick, or jump
3	Severe reaction: Moves side to side and tail flick, pronounced kick or jump, bawling, head shaking, or vocalization

**Table 2 animals-14-00255-t002:** High-Performance Liquid Chromatography (HPLC) specifications and run conditions.

Parameter	Details
HPLC	Agilent 1100 and 1200 Series
Column	ZORBAX Extend-C18; 4.6 × 150 mm; 3.5 µm
Mobile Phase	40:60—Acetonitrile:PBS, pH 7.4
Analysis Time	15 min
Flow Rate	1.0 mL/min
Injection Volume	10 µL
Column Temperature	28 °C
Detector	Agilent G1315B Diode Array Detector (DAD)
Wavelength	210 nm
Bandwidth	4 nm

## Data Availability

Data are unavailable due to privacy and proprietary reasons.

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
