# Peer review of "Assessment of the Pharmacokinetics and Pharmacodynamics of Injectable Lidocaine and a Lidocaine-Impregnated Latex Band for Castration and Tail Docking in Lambs"

_animals, 2024, doi:10.3390/ani14020255_

Round 1
Reviewer 1 Report
Comments and Suggestions for Authors
I question the relevance of the method for in vitro determination of lidocaine dissolution from the LLBs, and I am not convinced that this experiment adds value to the manuscript overall. Dissolution into PBS does not appear to be me to be a meaningful model of dissolution into the skin and subcutaneous tissues of a lamb. What is the relative solubility of lidocaine in the aqueous fraction of skin, compared to the oily fraction? Given the proximity of the ovine scrotum to the inguinal oil glands, does the aqueous:oil ratio vary between the scrotal neck and the tail? How relevant is the temperature of 35.1 degrees Celsius to the dermal temperature of a lamb?
Line 113. How many LLBs per basket?
Line 114; please express the centrifuge speed in relative centrifugal force (RCF) or gravities (g). A centrifuge speed described in rpm will result in different centrifugal force unique to the model of centrifuge and the rotor used, confounding any attempt by other researchers to repeat the procedure.
In line 374, ex vivo is italicised, whereas in line 376, in vivo is not italicised. Please be consistent.
Author Response
Reviewer 1:
I question the relevance of the method for in vitro determination of lidocaine dissolution from the LLBs, and I am not convinced that this experiment adds value to the manuscript overall. Dissolution into PBS does not appear to be me to be a meaningful model of dissolution into the skin and subcutaneous tissues of a lamb.
Indeed, the Dissolution test, per USP<711>, is an in vitro method for measuring release of an Active Pharmaceutical Ingredient (API) into an aqueous dissolution medium, and is not necessarily meant to accurately reflect the in vivo behaviour of a drug or medical device. Rather, it is used in the pharmaceutical industry as a QC test to ensure that the product is behaving consistently and to the desired specifications. We have included wording in the revised manuscript to clarify this (new lines 386-389) and that our intention with this test was to ensure the in vitro performance of the bands, per a standard test methodology, prior to testing lidocaine delivery in vivo (new lines 253-254).
What is the relative solubility of lidocaine in the aqueous fraction of skin, compared to the oily fraction? Given the proximity of the ovine scrotum to the inguinal oil glands, does the aqueous:oil ratio vary between the scrotal neck and the tail?
We believe that these questions are out of scope of the current manuscript, given the changes highlighted above.
How relevant is the temperature of 35.1 degrees Celsius to the dermal temperature of a lamb?
This number came from our previous study (see Saville et al 2020) measuring calf scrotums. We have since generated (currently unpublished) data indicating that the relevant tissues in lambs are between 35-37 degrees C.
Line 113. How many LLBs per basket?
Thank you for pointing this out. This has been clarified in the revised manuscript.
Line 114; please express the centrifuge speed in relative centrifugal force (RCF) or gravities (g). A centrifuge speed described in rpm will result in different centrifugal force unique to the model of centrifuge and the rotor used, confounding any attempt by other researchers to repeat the procedure.
RCF is not relevant here. A Dissolution Apparatus (per USP<711> and <1092>) is not a centrifuge; rather, it is a standardized instrument meant to accurately and precisely measure drug release into a liquid medium. A standard basket apparatus was employed in our study (i.e., standard dimensions, with volumes, temperatures, and stirring speeds all consistent with the aforementioned USP methods).
In line 374, ex vivo is italicised, whereas in line 376, in vivo is not italicised. Please be consistent.
Good catch, thank you. This has been corrected in the revised manuscript.

Reviewer 2 Report
Comments and Suggestions for Authors
The publication is interesting as pain relief in farm animals is a topic of
great importance, especially for routine mutilations such as tail docking and castration.
The complexity of this publication is that it investigates the effect of
LLBs via 4 different trials of which some relate to tail docking and
castration and others only to one of the two. While this is a strength of
the paper, it also makes it difficult to keep track of the animals involved
in the different trials and the precise experiments done. I also am missing information on the behavioural and stress-related effects on the animals during the experiments (e.g. when the biopsies were taken from control animals or from regions not sedated).
When reading the whole paper, I also do not feel the conclusion completely reflects the results seen and the results aimed at, namely to ensure that the animals have less stress and pain at castration/tail docking.
Lastly, while it is in the 'conflict of interest' part, I miss in the publication itself that several of the main authors work for the company having the patent on these LLB's. I would suggest adding this at least in a 'bias' part.
You can find some more comments in the document attached.

Author Response
Reviewer 2:
The publication is interesting as pain relief in farm animals is a topic of
great importance, especially for routine mutilations such as tail docking and castration.
The complexity of this publication is that it investigates the effect of LLBs via 4 different trials of which some relate to tail docking and castration and others only to one of the two. While this is a strength of the paper, it also makes it difficult to keep track of the animals involved in the different trials and the precise experiments done.
We agree—this is a complicated series of studies attempting to address a surprisingly complicated series of questions, and we have made our best attempt at clearly and concisely presenting them here.
We have described the studies in order in the Methods Section, which corresponds to the order of the Results Section and the data Figures; we have also attempted to be as clear as possible in the Figure legends as to the tissues, sample sizes, and experiments performed.
We would very much appreciate any suggestions that the reviewer may have to further simplify this article.
I also am missing information on the behavioural and stress-related effects on the animals during the experiments (e.g. when the biopsies were taken from control animals or from regions not sedated).
Good point. We did not thoroughly document behavioural or stress responses during the biopsy collection, as this was out of scope. The purpose of our study was to look at physiological levels of lidocaine versus a quantitative response to a controlled external stimulus, and to avoid the added complexity of potentially subjective assessments. With that said, we did not note major avoidance responses, even in the animals that received no lidocaine injection (or received control bands) prior to biopsy collection.
When reading the whole paper, I also do not feel the conclusion completely reflects the results seen and the results aimed at, namely to ensure that the animals have less stress and pain at castration/tail docking.
As stated in the Abstract and Introduction, “The objectives of this study were to assess the pharmacokinetics and pharmacodynamics of the current standard-of-care for pain mitigation in lambs during castration and tail docking (injectable lidocaine) and assess the ability of Lidocaine Loaded Bands (LLBs) to deliver therapeutic concentrations into the contacted tissues over time.” Assessment of pain and stress was not a stated goal of this work.
To this, we conclude “This study defined the effective concentrations of injectable lidocaine yielding 50% or 95% reductions in local sensation (EC50 and EC95, respectively). The use of injectable lidocaine provides effective short-term anesthesia for 120 to 180 min following the injection; however, additional strategies are needed to manage long-term pain. The use of an LLB could provide an alternative where tissue lidocaine concentrations meet or exceed the EC95 for at least 21-28 days and, based on electrostimulation data, provides local anesthesia for at least 3 days when compared to a control band…”
We do not think that our conclusions are out of line with either our data or our stated goals.
Lastly, while it is in the 'conflict of interest' part, I miss in the publication itself that several of the main authors work for the company having the patent on these LLB's. I would suggest adding this at least in a 'bias' part.
Field work was led by third parties (BR and AMH, who were masked as to the identity of control versus test bands) and data analysis was confirmed by a third party (SR).
We have clarified and expanded the COI statement per the editor’s suggestions, and we have also added that CCR is the patent assignee in the introduction (new line 92), per the reviewer’s comment in their annotated PDF (see below).
It appears that the reviewer is suggesting that an additional “bias” section be included elsewhere in the manuscript; we will defer to the Editorial Team as to whether this is appropriate, and how to best to execute it.
You can find some more comments in the document attached.
<animals-2748910-review>
We have replied to each of the reviewer’s comments in this PDF document, re-titled “animals-2748910-review_author response”.
